# "Can you do it for me?": Understanding Use-by-Proxy in Interactive Systems

Anon*
Anon

## ABSTRACT

"I can't reach", "my hands are full", "I'm driving"—can you do it for me? If using a smartphone is challenging for a user because of either physical or cognitive encumbrances, they often ask another person to perform the desired task on their behalf. In this situation, the user with the motivation or goal to perform the task is not directly using the device but is instead working through an intermediary, a *use-by-proxy*, where the proxy-user has limited initiative. Through a qualitative study, we probe this use-by-proxy phenomenon. We explore triggers, frequencies, and breakdowns that confound use-by-proxy interaction. We identify the challenges both for the end-user and the proxy-user (e.g., that proxy-user input is a deficient form of interaction for both the main user and the proxy user) and discuss consequences and implications for the design of this uneven collaborative interaction.

**Index Terms:** Project and People ManagementLife Cycle; 500 [Human-centered computing Interaction tech]: —

## 1 INTRODUCTION

As people become more attached to technology, we see an increase in users who struggle to operate their digital devices during a physically, socially, or cognitively taxing task. For example, a struggle to interact includes instances of smartphone use while physically encumbered [21, 22], while driving [33], or while having a conversation [38].

Human-Computer Interaction (HCI) designers have attempted to alleviate these challenges by designing alternative input methods to allow users to interact hands-free. Designers attempt to encourage hands-free interactions by refining of the form factor of digital assistants (e.g., Siri, Google Assistant, Alexa, Cortana); however, hands-free options have yet to replace the temptation to use direct interaction techniques [18, 45] even under conditions where use of a digital device increases the likelihood of injury [38]!

A simple work around often employed by over encumbered users is to request help from a friend to allow indirect interaction with the application through '*use-by-proxy*'. In this use case we can identify two main user roles: (1) primary user, who is motivated to interact with the application, and (2) the proxy-user who executes the task. For example, a passenger in a vehicle who assists the driver with navigation by using a Global Position System (GPS) application enters an address specified by the driver. We note that use-by-proxy can occur either on the main user's or the proxy-user's device.

Use-by-proxy is a collaborative interaction; however, the lack of parity in this collaboration creates an important niche user pair to study. In contrast to other forms of collaboration, during the proxy-user use case users are not motivated by the same goal; this is the main characteristic of use-by-proxy interaction. The primary user has a goal which necessitates the software use, and the proxy-user's goal is to assist. It is true—by its nature—that this characterisation

---

*e-mail: anon@email.com

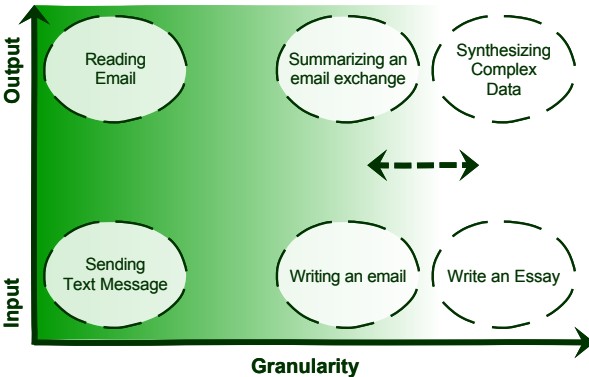

Figure 1: The design domain diagram above clarifies the scope of our topic represented by the shaded green area.

In this figure, we clarify the scope of our paper. The x-axis represents an increase in granularity of data. The y-axis indicates the direction of information as input or output. From the diagram, readers are informed that the scope of the research paper focuses on low granularity, simpler tasks e.g. sending a text message on behalf of a user is within scope, whereas synthesising a complex email conversation is not.

results in a range of possible interactions—from: a proxy-user replying to a text message by entering text via dictation; to the proxy-user writing a longer document for another user. A proxy-user can support input (entering an address into a GPS systems, for example) or output (providing directions from a GPS system to a driver).

In Figure 1, we describe this domain of use-by-proxy interactions and highlight (in green) our particular focus within this domain. We collect data on use-by-proxy scenarios that span the spectrum of use-by-proxy interactions with the goal of understanding the richness of these interactions.

Our paper contributes to the identification and understanding of the proxy-user edge use case, which results in an uneven collaboration between two users. Unique to this situation, the user who is interacting with the interface is not providing the main motivation to use the application. The primary user is instead accessing the functions of the application through a proxy, who we define as a secondary stakeholder.

To explore the proxy-user use case, we employ qualitative interviews, which probe participant experiences with use-by-proxy. Based on the results of our work, we highlight design implications and encourage designers to consider a proxy-use in computer-supported collaboration design—especially for applications commonly used while multitasking or in divided attention scenarios (e.g., driving, exercising, cooking).

## 2 BACKGROUND AND RELATED WORK

Literature on collaboration is expansive and includes exploration of multiple factors including, not limited to: time [19] or location synchrony [15], social learning [13, 17, 47], collaborator types [44, 48],

communication [41, 42], and territoriality [34, 36, 37, 46]. We specify our scope to be interactions between users engaging in uneven collaboration—specifically use-by-proxy interaction—because of the vast collection of work in this area, .

Literature on uneven interactions includes Parikh.2006's work [24] on mediated interaction. Mediated interaction is defined as a pattern of interaction where two or more users access the same device [24]. The 2006 report describes four kinds of inter-mediation: (1) cooperative scenarios, where users get (nearly) equal access to the interface, (2) dominated scenarios, where one or more users dominate the access, (3) inter-mediated interaction—which is necessary when a user has no direct access to a device but depends on the outcome— where users can see the output directly, and finally, (4) indirect interaction which deviates from its predecessor by removing the user's ability to observe the actions of their collaborator [24].

Restrictions in the physical space can prompt uneven collaboration. Shoulder-surfing content [4] is a well-studied aspect of public display use, which contributes to the staggered lineups of groups waiting for a display [3]. When shoulder-surfers begin to contribute to the interaction, we observe a use-by-proxy situation. Peltonen.2008 and Azad:2012:TBA:2317956.2318025 both investigated multi-user interactions with a touchscreen in a public space. The researchers concluded that the mediated interactions occurred because of the physical space limitations. Members of a large group gathered around the display behind limited subset of users, who could physically access the device. These physically distant users contributed to the interaction through proxy by advising and commenting on the user's actions [3, 25]. Similarly, studies of territoriality reveal that space constraints can stem from the division of collaborative spaces into territories [34, 35]. These territories limit access to the usable screen real estate.

Physical restrictions—as described above—limit access; however, physical constraints are only one factor that may bar users from interacting directly with an application. User's may be limited by their technology and computer literacy levels [31]. Sambasivan2010 investigated patterns emerging from different constraint constellations during intermediate interaction. They describe *surrogate usage* as input and interpretation of the output by a proxy-user, *proximate enabling*, which involves a proxy-user operating a device owned by technology-illiterate primary user, and *proximate translation* as system-output translation by proxy for the textual illiterate [31]. Furthermore, language proficiency and literacy levels can also be a barrier to use. Interaction may be carried out in an inter-mediated mode because of illiteracy of the primary user, fear of technology, habits of dependency, costs or access constraints such as age [31]. Use-by-proxy clearly has accessibility advantages.

In contrast, collaboration may be sought based on the value of the contributor. For example, expert knowledge in domestic IT infrastructure is a contributing reason for the frequency of proxy-user input scenarios. Kiesler.2000, reports that the intellectual authority in families can be shifted when a skilled or motivated member (mostly—but not necessarily—a teenager [5]) becomes the "family guru" [16]. Poole.2009 were motivated to explore factors which influence the way such "helpers" provide aid. The paper found although helpers often did not advertise their skills, they actively maintain their identity as experts and get frustrated when presented with an unsolvable problem or when a distrustful person requests help [26].

Proxy-user computing is a common input technique in operating rooms [10] and is referred to using different terms in the medical literature including: *task delegation* [11], *assistant-controlled computer keyboard* [49], *assistant-in-the-middle* [8], or *yell-and-click* [43]. Surgeons have limited access to computing input devices because of the need for complete sterilisation [20, 29, 43].

Information handover in medical spaces is critical [14]. Although surgeons may rely on the information output by a computer appli-cation, they—quite literally—have their hands full [23]. Therefore, verbal delegation of computing tasks to a proxy-user is common in these environments [20, 43]. Proxy-user collaboration is often considered a benchmark for surgeon-computer interaction [11, 27, 49]. As a result, the compendium of medical literature provides insight into proxy-use situations and exposes the delicate nature of communication between a primary user and a proxy-user.

In 2004, Grange.2004 published a case study where a misunderstanding between the surgeon and the proxy-user resulted in error. In an attempt to recover from this error, hospital management allowed intervention of three additional assistants. Despite the added resources, the final resolution of the problem required the surgeon to halt the surgery, remove themselves from the sterile environment, and access the computer directly. The actions of the surgeon resulted in a delay of eight minutes. An eight-minute delay could be fatal because of the critical nature of surgery [8].

The disadvantages introduced by proxy-user scenarios are clarified explicitly in the above example: Proxy-use is prone to misinterpretation, is indirect, and depends on the assistants' experience level. Therefore, research into who is selected as an assistant becomes relevant. Selecting a proxy-user goes beyond surface level attributes (e.g., race or gender). Instead, scholastic abilities, aptitude, extroversion, and high participation levels dictate the desirability of a potential proxy user [12]. The reported selection criteria [12] are in line with classic research on collaboration in the children's playroom by Cockburn.1996. These researchers reported that collaboration among children benefited from any kind of negotiation. Mutual awareness and breakdowns even occurred in successful collaborations, while domination or ignorance indicated less effective situations [6].

Selection of collaborators or requests for help in the workplace further reveals reasoning behind collaborator selection. Adams.2005 reported on how different methods of accessing digital libraries are perceived in academic and health-care institutions. Digital information was made accessible (1) via existing computers in people's offices and libraries, (2) shared spaces, and (3) by information intermediaries supporting the users (i.e., clinical staff). Users in academia using personal computers report few points of contact with librarians and criticised the library system. In contrast, medical professionals working with computers in the hospital ward, expressed that they felt a lack of personal competence, which was exacerbated when asking a younger colleague for support. Information intermediaries—which act as an interface between clinical staff and the digital library—add librarian domain knowledge; they were seen as beneficial for effective information usage [1].

After a review of the literature, it becomes apparent that use-by-proxy is an area of application design that warrants further exploration.

## 3 METHODOLOGY

To explore use-by-proxy interaction, we use a qualitative approach to investigate the occurrence of a proxy-user situations. The overall goal of our study is to better understand participants' expectations and experiences of use-by-proxy. We investigated tasks in which one user is the main motivating driver (i.e., proxy-user scenarios).

For our study, we define the proxy-user use case as: *a task where the primary user, who is motivated to operate the system, asks another human user to interact with the system on their behalf. The recruited user assisting by allowing the primary user to use the application by proxy, is what we have identified as the proxy-user*. For example, when driving a car, the driver may ask the passenger to get directions from a mapping application.

### 3.1 Participants

Eighteen adult participants (18+ years old) were recruited via mailing lists and took part in individual interviews (described below).

They were remunerated with 20€.

## 3.2 Interview Structure

To understand and identify the definition of the proxy-user, our interviews began by asking participants about situations in which use-by-proxy may have occurred. To clarify use-by-proxy, we provided a number of scenarios to motivate discussion:

- Driving with a navigator,

- cooking together,

- putting together furniture with a friend,

- fixing a bike together,

- pair programming, and

- working collaboratively in their school or work career.

Additionally, we also asked how the relationship to the other user (friends, family) or the environment (who owns the house or car) affected the situation.

We encouraged participants to structure their description of use-by-proxy as a walkthrough of their interactions with or as a proxy-user in various situations they identified. In our interviews, we attempt to elicit what is different in proxy-user collaborations. We ask the interviewees for insight into their thoughts, feelings, and decisions in these situations. Our goal was to identify factors that contribute to the division of responsibilities, including the unevenness of skill, and the vulnerability of asking for help. We also discuss breakdowns in collaboration, such as providing too much or too little help. Finally, we look at outside factors that influence the relationship between collaborators.

### 3.2.1 Qualitative Interview Analysis

The qualitative data was analysed in accordance with the procedures outlined by Corbin, Strauss, et al. [7, 39, 40]. To analyse the data, quotes were separated from general discussion (e.g., quotes regarding opinions vs. introductory conversation). Using a bottom-up approach, the quotations were aggregated and sorted using an affinity diagramming technique. Next, aggregated data points were analysed to pull relevant ideas and information. Overlapping categories were then explored using top-down analysis based on the themes arising from the data. Afterwards, related clusters of themes were analysed to uncover detailed differences, identify overlapping concepts, and pull larger higher level concepts from the data. The resulting work comprises the themes presented in the paper.

## 4 RESULTS

Using the qualitative methodology outlined above, we present our results explaining *what* a proxy-user is, understanding *why* proxy-users are helpful, and finally, *how* to engage with proxy-users and navigate the interaction.

## 4.1 Proxy-User: an Uneven Collaborator

As we note in our introduction, use-by-proxy is an uneven collaboration. This observation that a proxy user was an uneven collaborator was supported by our data. For example, because the main motivation for interacting with the application is central to the primary user, the primary user also is the most invested in the outcome.

> "You have less responsibility. You have maximum responsibility for your task, for a special task you're doing or for part of the result, but I think the leader has a responsibility for everything that's happening. Also, for things other people do, and maybe don't do very good. So he's having more responsibility with more risk that if something goes wrong it's his fault" P1

Participants felt that, despite the role of the proxy-user being more akin to an assistant, the work is still a contribution and should be treated with the same respect as expected from any collaborative arrangement. Participants expressed that expectations of fair collaboration, positive leadership, and teamwork still apply.

> "I think that's difficult if you're just the assistant and have to follow orders or the ideas of someone else, but still, if you have the feeling that you're contributing something valuable or something important, I think it can be a positive experience all in all. On the other hand, I think it also can be frustrating if you're not valued; only do the back work or not necessary stuff." P1

Moreover, proxy-user interactions remain susceptible to the pitfalls of any collaboration because finding a good collaborator is still a challenge. A collaborator can be unreliable, make mistakes, or misunderstand instructions entirely.

## 4.2 Help: The Great Trade-Off

Choosing to collaborate with a proxy-user can have both negative and positive benefits as a result of the ever present differences between computers who reliably execute a task with consistency and humans who reliably introduce variability.

> "I think it's just small errors, small human errors, like sometimes people do mistakes, and most of the time it's working right, but sometimes you have these inaccuracies in the description... For example, 'Okay, you have to take the third right.' Like, I don't know if they mean only the main streets, or the small streets in between, if you have blocks, you know? Like, do they count this small entry, like this small street, as a turn left, or do they only mean like the big junctions, you know? Like that kind of thing." P4

One challenge with proxy use, and risk to the utility of proxy users, is that humans can be unpredictable when it comes to the delivery of information. Anything from the clarity, quantity, depth, or framing of the information can vary. Thus the trade-off between human vs. computer help is especially salient when comparing a human's interpretation to an expected computer-derived outcome. Since the primary user is expecting use-by-proxy to the application, the variability in human delivery can cause conflict. P4 explains:

> "Like people usually don't guide me wrong, but in these scenarios, like I said, where there's margin for error, I'd rather trust the map. But I said, I don't trust the map, but I trust myself to interpret it right." P4

As this quote illustrates, the primary user is dependant on the proxy-user's interpretation of the information. The proxy user is using an application and providing a synthesis of that information, a situation which can be irritating to a primary user who feels that they could have out performed their proxy-user assistant. For proxy-user interactions to be successful, the primary user must trust that the proxy-user is capable of completing the task and outputting the correct information.

The variability in human communication presents positive benefits as well. Computers are limited in both the information that they accept and provide, and are also unsuited for particular types of information (e.g. emotional or expressive statements). Additionally, a proxy-user is able to rephrase, verify, and correct actions in accordance to the direction of the primary user.

> "I can communicate when there are other problems, for example, the real situation is always different than the situation on Google Maps. And if I have a person next to me, I can say, I don't understand what you mean, and can you explain it again. I think it's better." P8

Moreover, the primary user does not need to worry about the format in which they input information to a human proxy-user.

> "It's easier and quicker to tell them, 'Hey! Google that!' I mean, you can describe things and you don't have to think about how to Google it. You can just describe stuff, like a building, and tell them: 'hey find out what this is'. And you can concentrate on driving." P12

In summary, a proxy-user can also provide rich information that is customised to whom they are talking to. Alternatively, a proxy-user can also simplify and filter out unnecessary information. The overall effect of these decisions can result in a tailored experience for the primary user in real time, but the challenge with giving and receiving help is in matching expectations. Essentially, because in some instances the primary user wants information to guide their judgement, and in other instances they desire synthesis and judgement to simplify information sharing, interacting through or as a proxy-user presents pitfalls that are difficult to navigate.

### 4.3 Navigating the Proxy-User Relationship

One significant factor that influences proxy user relationships is the relationship between primary and proxy user. The better a proxy-user knows the primary user, the more additional cues – based on this pre-existing relationship – can be used to enhance proxy use. For example, a participant discussed how the close personal relationship with their mother results in better navigation information because the mother acting as a proxy-user will provide more information based on the primary user's emotional cues.

> "I think because my mom can tell how I feel. 'She looks nervous, I have to tell her what comes next'." P2

Participants frequently noted that a closer relationships may result in better communication, or the relationship may act as a rapport for the trustworthiness of information.

> "I think that lots of factors. The relationship to the person you are trying to help or helping you. I mean I'm ... Everything is different, you know, when I'm trying to help my parents in language or whatever than helping for example my girlfriend or some student or some friend. I mean that's all different. The kind or the type of relationship you have, and age maybe too. Yeah, we'll talk differently to my grandma than to my mother, for example." P6

Pre-existing relationships can also result in a more positive or fun experience in and of itself, particularly through joint struggles.

> "It depends who the person is, but if it's a friend then it's, I think, alone a positive experience to interact with one another, even if it's just finding the way. I think this positive connection or interaction doesn't happen if you just have your phone, even if the phone is telling you, or Siri is telling you where to go. I think this positive experience is lacking." P1

That being said, one challenge with pre-existing relationships is that proxy-user interaction can also be more volatile. Close connection provides the possibility for expressing dissatisfaction, whereas more distant relationships are less likely to experience this tension.

## 5 DISCUSSION

Our investigation illustrates challenges associated with the uneven collaboration between the primary user and an assisting secondary user who enables use-by-proxy of an application.

In a proxy-use collaboration, two users look towards an application with two different underlying motivations and expectations.

In motivation, the proxy-user is altruistically motivated to help and expects that the experience will be generally positive and socially rewarding. In contrast, the primary user who is seeking assistance enters the interaction with expectations that the proxy-user will preform at least on-par with the application in the current situation. Extending this to expectations, despite positive intentions motivating the proxy-user, the introduction of a person-in-the-middle does not always alleviate the burden placed on the primary user. Instead, the primary user may become frustrated due to differences in the communication of application outcomes. Given that many people are motivated by their own needs for competence and autonomy [30], shifting the control to proxy-user can have a demotivating effect for the primary user. The proxy-user who expects positive social collaboration, respectful guidance, and feedback may reflect back negativity, resulting in conflict between both stakeholders.

Our qualitative approach reveals that due to the differences in motivation and expected outcomes, the proxy-user use case differs from the crafted UX designed for the application. Therefore proxy-users are a challenge to designers whom typically focus on creating a usable interface for a single dedicated, directly motivated user.

Identifying the proxy-user use case demands further thought into the design of an interface, especially for applications designed to offer assistance during the cumbersome tasks that also motivate a request for use-by-proxy (e.g. driving or cooking). Our results highlight that the proxy-user may be skilled at the task, but has limited liability to the outcome and may need additional guidance.

Both users in the proxy use case face challenges completing distributed responsibilities. These challenges are further exacerbated by the need to manage the overlaid social challenges accompanying any collaboration. Moreover, given that the role of proxy-user exists to help others, one question we can pose is: beyond altruism, what encourages the user to maintain the collaboration? The question becomes especially relevant when the proxy-user is forced to maintain the role (i.e. over long term tasks). Motivated by the challenges of proxy use, we pose a series of design questions that seek to present alternatives to better support this uneven collaboration.

### 5.1 Question 1: Can we eliminate the proxy-user?

Motivated by the notion that this form of collaboration is undesirable, designers may wish to negate the use for a proxy. To accomplish this, designers must overcome shortcomings of the current design. The question then becomes how best to do this.

Understanding the workarounds of a current system can help us determine what the system should do [9, 32]. In the proxy-user use case, demands made on the proxy-user indicate design directions by allowing designers to understand the gaps in the application's current design and implemented features. We argue that, if it is conceivable that an application could require use-by-proxy (and many may feasibly be used this way), user testing protocols should include a proxy-use scenario to understand the different and changing expectations of both primary and proxy-users as they complete demanding tasks.

In many situations, designers already, in part, propose solutions designed to alleviate the need for proxy use. For example, designers of applications geared to multitasking or divided attention scenarios (e.g. driving, exercising, cooking) have explored multiple methods to increase hands-free capabilities. For example, many companies are exploring smart assistance, smart homes, and voice activation. Given both progress and desired system features [45] the elimination of proxy use may be feasible in certain contexts. Realistically, the existence of the proxy use case indicates that advancement in hands-free and smart assistant technology still cannot replace the desire to ask for help from another human. However, if designers actively explore these use-cases, work to understand heterogeneous expectation of information and synthesis, and continue to explore alternative designs, it seems feasible that technological advances can

begin to partially address the proxy use-case by, in whole or in part, eliminating proxy-users.

## 5.2 Question 2: Can we re-balance the collaboration?

As an alternative approach, designers may attempt to balance the uneven collaboration of the proxy-user by shifting the experience profile from *Assistance* to *Collaboration*. The goal would be to transform an "over-the-shoulder-boss" to a more equitable "pair-programming" paradigm.

We can support a re-balance of collaboration by supporting the fundamentals of proxy-user collaboration: communication, attitude, and skill. At a low-level this may include adding gaze-level support to understanding how primary user may attempt to gleam information from the application as it is operated by the proxy-user. Alternatively, to help a proxy-user comprehend the instructions given, designers may consider creating a wizard or workflow that a proxy-user can follow to distil all the necessary information from the application to the primary user with concise, easy-to-follow language. These distillations can be geared toward different levels of abstraction: information, synthesis, suggestions, and alternatives. Moreover, a new simple visualisation mode could be added with the intention of supporting the proxy-user's explanation. For example, simplistic and blurred visuals may supplement the proxy-user's instructions and reduce risk of distraction by presenting limited visual information.

At a higher level, we want our tools to contribute to the distributed or shared cognition [2, 28] necessary for a successful proxy-user interaction. By shifting responsibilities away from the primary user completely, we may obligate the division of cognition between two users. Strategies can include designing an overview for the proxy-user instead of tailoring the interaction directly towards the primary user. For example, instead of providing an overview of information, allow the proxy-user to build a custom notebook themselves using the application's tool set. Additional functionally can support the proxy-user, who may flip ahead in a manual or along a route and attempt to create a mental summary. While successful recall of information is difficult and learning takes time, if proxy-users can pre-learn, they may become more informed collaborators.

## 5.3 Question 3: Can we aim to guide the proxy-User

An ongoing challenge for applications is that they are currently designed for a single user, the device owner. However, there are a number of applications where use-by-proxy seems obvious, including navigation systems. Instead of targeting the primary user, the application could target the proxy-user in a secondary mode by changing the presentation of information from a workflow aimed at the person completing the task to a workflow for a person who seeks to assist.

As in the previous example of re-balancing, a modification of the presentation of information can focus on queuing information to help the proxy-user prepare and anticipate next steps. For example, the application could identify larger time gaps between steps to indicate a good time to communicate complex information to the primary user. Moreover, designers might employ picture-within-picture views to allow for peripheral monitoring by the proxy-user during these longer inactive time periods. Notifications of upcoming events can help avoid the sudden call-to-action experienced by a proxy-user while multitasking.

## 6 CONCLUSION

It is immediately obvious to anyone who navigated, transcribed a text message, followed a recipe, assembled furniture, or controlled a slide show that, in these situations, sometimes another person will provide information – often from an application – while the user performs a physically or cognitively challenging task. We label these non-primary users 'proxy-users'. In exploring the HCI and Design research literature, design explorations seem almost silent on an analysis of proxy use and the proxy-user experience. What happens in proxy use? Are there pitfalls? Are there opportunities for improvement?

Our paper contributes to this design space by highlighting areas for further investigation into this unique form of uneven interaction. Can it be eliminated? Re-balanced to more equal collaboration? Enhanced to improve proxy use efficacy? This paper seeks to formulate these basic questions to encourage more targeted discussion of this unequal, yet frequent, style of collaborative application use.

## ACKNOWLEDGMENTS

The authors wish to thank A, B, C. This work was supported in part by a grant from XYZ.

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
