# OpenReview forum: "``Can you do it for me?'':  Understanding Use-by-Proxy in Interactive Systems"
_graphicsinterface.org/Graphics_Interface/2023/Conference — Submitted to GI 2023_

### Official Review · Reviewer_zNq8 · 2022-12-29
**This paper presents the findings of a qualitative interview study on a common phenomenon in everyday life, use-of-proxy, to identify expectations and experiences of the parties involved and the associated barriers, and to propose design implications. The topic is interesting but the study is too broad, thus a proper scoping would benefit it. In general, there are major flaws in the paper/study.**

**Rating:** 3
**Confidence:** 4

**Review:**

The authors tried to use surgeon-proxy scenario in the Background to amplify the significance of effective use of proxy and how a lack of it could lead to critical consequences including death. However, it is unclear if such practices in an operating room are all use-of-proxy and not simply work collaboration with varying responsibilities. Also, delay in day-to-day proxy-use such as during driving or cooking would usually result in more time for the users to complete the tasks, without causing serious consequences.

For the study, “Eighteen adult participants (18+ years old) were recruited via mailing lists”. But it is unclear what mailing lists were used, e.g., mailing lists of social circle, students, employees, etc. This information could impact the participants’ demographics which are entirely missing, other than being 18+ years old. Their average age/age ranges, education, occupation, technology experience, etc. could help contextualize the reported findings. Also, since the study aimed to find out what a proxy-user is from the participants’ perspectives, it’s important to describe how the question was framed around the use-by-proxy scenario and what a proxy-user is, and whether the definition of a proxy-user use case stated in the paper was mentioned to the participants. It’s also unclear how many use-or-proxy scenarios were described and what role the participants played. It seems the authors have made the assumptions that the proxy-user is always a helper/assistant who is not motivated to operation a system s/he is not familiar with, rather than one who could be as motivated and proficient with system as the primary user. For example, when two people go on a road trip to the same destination, they may be equally motivated and familiar with the GPS system. Alternatively, the proxy-user may even be more motivated and more familiar with the system than the primary user who may only be tasked with the driving responsibility. Thus, it seems this study has pre-determined the collaboration being a master/helper dichotomy with uneven collaboration, making the findings on “what a proxy-user is” not useful.

Moreover, the type of interviews conducted was not stated, whether it’s structured, semi-structured, unstructured, and how long the interviews last which could imply the depth of the investigations? Were the interviews transcribed before analysis or field notes were analyzed instead?

The lack of (demographic) information of the participants and the corresponding proxy-user/primary user, and the description of the specific use-case/application at hand makes the interpretation of findings, particularly the quotes, difficult. It’s confusing that the paper stated “the application” in the findings and the discussions, implying a specific application was used by all the participants, which however was not the case.

While the use of verbatim quotes can help contextualize the findings, I find those in this paper not in alignment with the reported findings. For example, it’s unclear how the quotes by P4 in 4.2 relate to said “trade-offs”. What described in the quotes can be found in any kind of human-human communication, not just use-of-proxy scenarios. As also mentioned later in the same subsection, people can easily ask for clarification or elaboration in real time whereas there’s no easy way for human-computer communication, especially when using conversational user interfaces. So is this trade-off between human-human communication and human-machine communication, instead of in primary/proxy user communication?

What does it mean by “One significant factor that influences proxy user relationships is the relationship between primary and proxy user”? What is proxy user relationships? Did the authors mean proxy use efficacy instead?

“In a proxy-use collaboration, two users look towards an application with two different underlying motivations and expectations.” This is stated such as-a-matter-of-factly that it’s important to know if this is the authors’ own opinion/assumption or from the literature (citation is needed then).

“Our qualitative approach reveals that due to the differences in motivation and expected outcomes, the proxy-user use case differs from the crafted UX designed for the application.” This is confusing as the proxy-user use case (only one?) was defined as a task so how can a task be compared with the crafted UX?

For the discussion, I find the question “can we eliminate the proxy-user?” unrealistic. Say in the same driving scenarios, many people drive alone, without a proxy-user so proxy-user is not always necessary. Also, think back to the time when people used printed maps. People would print out maps and detailed directions before road trip, again without any proxy-user. In situations like those in operating rooms as already described above, the personnel involved should have been trained to (using specific systems) perform different tasks and responsibilities. Thus such situations may not present use-of-proxy use cases, but a division-of-labor collaboration. Despite that, the so-called “proxy-users” are typically more proficient with the system at hand than the primary-users. Finally, the authors may want to explore whether the use-of-proxy should be considered as a spectrum, instead of a dichotomy.

---

### Official Review · Reviewer_nvch · 2023-01-11
**A good explorative user study about the special interaction condition of having another human to become their proxy.**

**Rating:** 4
**Confidence:** 3

**Review:**

The paper discusses how use-by-proxy, a condition where primary users ask the proxy users to perform interactions as they will. The authors explored using qualitative user interviews when this condition triggers, how often this triggers, and what the sub-procedures to the proxy interactions exist.

In general, the paper provided a sound definition of the proxy-user use case, which can be a great start to the exploration, and the paper is well-written to describe the crucial points they explored. Also, the results and discussion seem to open up good directions for how future research on this topic can take. Below are some of my comments:
- The authors posed a question, to see if we can potentially eliminate the proxy user. As per the interactions described by the authors, there can be technological solutions such as hands-free voice assistants or some smart quickstart buttons, etc. However, it would be great to include why humans still desire to ask for help from another human. Not only the functionality and usability matter but also the reliability and building trust based on the robust actions from those available functionalities can also play a psychological role in the bias towards choosing human-proxy. In short, it would be great if this proxy request behaviour could be paired with the measure of trust from smart interactions. Not only asking the passenger to get directions but to an extent, assuming there exists a fully autonomous vehicle available, would humans still ask their friends/family to drive them to another location? What about asking a stranger to help them with certain functionalities that they cannot perform at the moment? As the paper explored, the relationship is also related to perceived reliability or trustworthiness.

- It seems like the trade-off is related to the weight of responsibility that the proxy user feels. Can there be a point where the proxy user declines to be a proxy user?

- Please elaborate further about the details in the interview structure and questions, etc.

- The demographic information is missing. Even if it did not play the role in the analysis, for the readers and audiences, the information is a crucial lens to understanding the samples.

- Statements that are based on assumptions need references, otherwise from the interview. The ground theory approach should present general codes with considerable overlapping ideas at least as a table or a figure.

---

### Official Review · Reviewer_rTJB · 2023-01-13
**Qualitative study - Use-by-Proxy**

**Rating:** 3
**Confidence:** 4

**Review:**

The paper investigated the “use by proxy” phenomena, creating an unbalanced collaboration dynamic. While I see value in exploring this concept for designing collaborative technologies, the paper does not clearly states the use of such a concept in tech design (HCI). The motivation for the presented study is missing.
An interview with eighteen participants was conducted to better understand the phenomenon. There is limited information provided about the participants; have these participants had an experience using a proxy technology? What is their technology literacy?
The scenarios given to participants are very different and some do not have any technology involved. The questions asked in the interviews are not included in the study description, which makes evaluating the results very difficult. Additionally, I am concerned that the study questions were too broad and that each scenario could be a study by itself when discussed in detail.
The results of the study are too general and it is rather difficult to map these results to a specific technology. The context of the discussion is missing in the results and participants’ quotes. I recommend focusing on a specific scenario, such as assisting with driving (GPS), targeting the study on this scenario, and analyzing the results to draw design guidelines for the chosen scenario. This can be beneficial for designing collaborative tools in the chosen scenario.
Overall, I find the concept that is studied very interesting; however, the study design and results do not fully answer the research questions posed. Additionally, the writing of the paper needs to be edited and revised. I do not think the paper is ready for publication at this stage.
Minor edits:
One paragraph in the introduction is separated from the section; I suggest addressing this separation.
There are several grammar mistakes in the text (many in the Related work) that need to be fixed.

---

### Meta-Review · Area_Chair_ZSeP · 2023-01-16

**Recommendation:** 3
**Confidence:** 4

**Metareview:**

The paper defined and investigated proxy-user using a qualitative study.
Reviewers found the paper a great start to the exploration of proxy-user concept. They found value in exploring this concept for designing collaborative technologies.

Reviewers had some concerns. A reviewer asked that the proxy request behaviour be paired with the measure of trust from smart interactions. Another reviewer was unsure if the authors assumed a helper role for the proxy-user or if other roles were also considered. Lastly, the study description lacks details such as participants’ demographics, method of recruitment, etc.

Reviewers’ recommendation in summary:
-	Clearly state the context that the proxy-user has been studied, the intent of the study, and how it will be useful for future designs.
-	I suggest the authors add the necessary details requested by reviewers about the study in the paper.
-	Clearly indicate the role and the weight of responsibility that the proxy user takes vs the user.